# Chrysin Induces Apoptosis via the MAPK Pathway and Regulates ERK/mTOR-Mediated Autophagy in MC-3 Cells

**DOI:** 10.3390/ijms232415747

**Published:** 2022-12-12

**Authors:** Gi-Hwan Jung, Jae-Han Lee, So-Hee Han, Joong-Seok Woo, Eun-Young Choi, Su-Ji Jeon, Eun-Ji Han, Soo-Hyun Jung, Young-Seok Park, Byung-Kwon Park, Byeong-Soo Kim, Sang-Ki Kim, Ji-Youn Jung

**Affiliations:** Department of Companion and Laboratory Animal Science, Kongju National University, Daehak-ro, Yesan-gun 32439, Chungcheongnam-do, Republic of Korea

**Keywords:** chrysin, apoptosis, autophagy, mucoepidermoid carcinoma, oral cancer

## Abstract

Chrysin is a flavonoid found abundantly in substances, such as honey and phytochemicals, and is known to exhibit anticancer effects against various cancer cells. Nevertheless, the anticancer effect of chrysin against oral cancer has not yet been verified. Furthermore, the mechanism underlying autophagy is yet to be clearly elucidated. Thus, this study investigated chrysin-mediated apoptosis and autophagy in human mucoepidermoid carcinoma (MC-3) cells. The change in MC-3 cell viability was examined using a 3-(4,5-dimethylthiazolyl-2)-2, 5-diphenyltetrazolium bromide cell viability assay, as well as 40,6-diamidino-2-phenylindole, annexin V, and propidium iodide staining. Western blotting was used to analyze the proteins related to apoptosis and the mitogen-activated protein kinase (MAPK) pathway. In addition, the presence or absence of autophagy and changes in the expression of related proteins were investigated using acridine orange staining and Western blot. The results suggested that chrysin induced apoptosis and autophagy in MC-3 oral cancer cells via the MAPK/extracellular signal-regulated kinase pathway. Moreover, the induced autophagy exerted a cytoprotective effect against apoptosis. Thus, the further reduced cell viability due to autophagy as well as apoptosis induction highlight therapeutic potential of chrysin for oral cancer.

## 1. Introduction

Mucoepidermoid carcinoma (MEC) is the most common type of malignant tumor observed in the salivary glands, accounting for 3–12% of all cases. The available treatments include surgery and chemotherapy, whose potential side effects could induce a high level of stress in patients [1,2]. Therefore, the focus on treatments based on natural substances that could reduce the side effects of chemotherapy has notably increased [3,4].

Flavonoids are generally found in fruits and vegetables as naturally occurring polyphenolic phytochemicals classified into flavanols, flavanes, and others [5]. Chrysin (5,7-dihydroxyflavone) is a natural flavonoid enriched in various plant extracts, as well as honey, which has demonstrated anti-inflammatory, antioxidant, and anticancer effects against various cancer types [6,7,8]. Nonetheless, the anticancer effect of chrysin in oral cancer cells and the related mechanisms are yet to be identified.

Apoptosis is a natural mechanism of programmed cell death. Apoptosis controls cell growth and aging and plays a critical role in cellular development and homeostasis [9]. The intrinsic pathway of apoptosis involves mitochondria, as it occurs through changes in the ratio of B-cell lymphoma-2 (Bcl-2) proteins, including pro-apoptotic proteins, pro-apoptotic BH3-only proteins, and anti-apoptotic Bcl-2 proteins [10].

Autophagy is a ‘self-eating’ process, where large domains, such as the cytosol and cellular organelles, are engulfed by autophagosomes and then transferred to lysosomes for degradation. Through this process, cells obtain energy sources, such as amino acids and sugars, and maintain homeostasis under stress conditions [11,12]. However, this is also referred to as a ‘double-edged sword’, because when autophagy becomes extensive and extends beyond homeostatic functions, it can induce cell death. Certain recently identified drugs harness this effect to control autophagy and exhibit anticancer effects [13,14].

In stress-induced autophagy, protein light chain 3 (LC3) is converted into esterified LC3II. The coiled-coil myosin-like B-cell lymphoma-2-interacting protein (*Beclin-1*) is a crucial gene that locates other autophagy proteins at the pre-autophagosome site. Hence, LC3 and Beclin-1 are regarded as autophagy markers [15]. The mammalian target of rapamycin (mTOR) is a key regulator of autophagy regulated via the mitogen-activated protein kinase (MAPK)/extracellular signal-regulated kinase (ERK) pathway. When induced by the MAPK/ERK pathway, mTOR prevents excessive autophagy and affects cancer cell survival [16,17].

The MAPK pathway plays an important role in cancer progression and development. It consists of ERK, c-Jun N-terminal kinase (JNK), stress-activated protein kinase (SAPK), and p38 kinase MAPK [18]. While ERK1/2 is involved in cell survival and proliferation, JNK and p38 respond to a variety of stress factors, such as ultraviolet light, oxidation, and DNA damage [19]. Once activated, a crucial role of the MAPK pathway is to convert extracellular stimuli into various cellular responses, such as differentiation, proliferation, and aging [20]. In this study, chrysin induced apoptosis and autophagy via the MAPK/ERK pathway in MC-3 oral cancer cells. Furthermore, we investigated autophagy directionality in oral cancer cells and its association with apoptosis.

## 2. Results

### 2.1. Chrysin Reduces the Viability of MC-3 Oral Cancer Cells

After 24 h treatment of MC-3 cells with chrysin (Figure 1A), a 3-(4,5-Dimethylthiazol-2-yl)-2,5-diphenyltetrazolium bromide (MTT) assay was performed. Compared with control-treated cells, chrysin-treated cells had a significantly lower viability after 25 µM (Figure 1B). Based on this, chrysin concentrations of 0, 50, and 100 µM were used in subsequent experiments.

### 2.2. Chrysin INDUCES Morphological Changes in MC-3 Cells

Following 24 h treatment of MC-3 cells with chrysin, 4′,6-diamidino-2-phenylindole (DAPI) staining was performed, and through the process of nuclear condensation and shrinkage, apoptotic bodies responding to the reagent were counted, quantified, and analyzed. Treatment with 0, 50, and 100 µM of chrysin showed apoptosis rates of 0.3, 3.0, and 9.0%, respectively, showing a significant increase compared with the rates of the control (Figure 2A,B).

### 2.3. Chrysin Induces Apoptosis in MC-3 Cells

Following 24 h treatment of MC-3 cells with chrysin, annexin V/propidium iodide (PI) staining and subsequent flow cytometry were performed. The apoptosis rate was measured as the percentage of annexin V-positive cells. Treatment with 0, 50, and 100 µM of chrysin showed respective scores of 6.6, 12.5, and 16.5%, showing a concentration-dependent increase in apoptosis rate (Figure 2C,D). Western blotting was performed to identify apoptosis-related proteins. As a result of Western blotting, the pro-apoptosis proteins c-PARP and Bax showed an increased tendency, whereas the anti-apoptosis protein Bcl-2 decresed (Figure 2E).

### 2.4. Chrysin-Induced Autophagy and Association with Apoptosis in Oral Cancer Cells

Acridine orange staining of MC-3 cells after treatment with chrysin showed that acidic vesicular organelles (AVOs) increased in a concentration-dependent manner (Figure 3A). In addition, Western blotting showed that the levels of autophagy related proteins LC3-II and Beclin-1 increased, whereas those of p62 and p-mTOR decreased (Figure 3B). Suppressing chrysin-induced autophagy using 3-methyladenine (3-MA) and hydroxychloroquine (HCQ), commonly used autophagy inhibitors, showed that cell viability did not vary significantly between the chrysin-only and chrysin + 3-MA groups. In contrast, the chrysin + HCQ group displayed further reduction in cell viability, compared with cell viability in the chrysin-only group (Figure 4A). The changes in apoptosis rates and the related proteins after treatment with HCQ showed that the rate of apoptosis and the level of Bax/Bcl-2 increased considerably in the chrysin and HCQ groups, compared with those in the chrysin-only group (Figure 4B,C).

### 2.5. Chrysin Induces MAPK/ERK-Mediated Apoptosis and Autophagy in MC-3 Cells

Chrysin-induced changes were examined in ERK1/2, JNK, and p38, key proteins of the MAPK pathway. With an increase in concentration, p-ERK1/2 decreased, whereas p-JNK and p-p38 increased (Figure 5). In addition, cell viability was reduced to a greater degree in the group simultaneously treated with PD98059 and chrysin, compared with the group treated with chrysin alone (Figure 6A). Furthermore, Bax/Bcl2 and Beclin-1 levels increased, but p-mTOR levels decreased (Figure 6B).

## 3. Discussion

Chrysin is a flavonoid abundantly found in honey and plants, with anticancer effects against various cancer types. Previous studies have shown that chrysin reduces the viability of prostate and gastric cancer cells [21,22]. These results coincide with the MTT assay results in this study, collectively suggesting that chrysin can reduce cell viability in various cancer types.

When apoptosis is induced, cytosol and DNA condensation leads to the formation of apoptotic bodies [23]. In addition, pores form on cell membranes to expose phosphatidylserine [24]. These features were examined using DAPI and annexin V staining, and the percentage of annexin V-positive cells and apoptotic bodies in the chrysin treatment group, was higher than that of the control. According to a previous study, chrysin increased the number of apoptotic bodies and induced apoptosis in lung cancer cells [25]. These data support our study and suggest that chrysin could reduce MC-3 cell viability through apoptosis. The intrinsic apoptosis pathway is triggered by changes in the Bax/Bcl-2 ratio, frequently detected as markers of apoptosis alongside c-poly (ADP-ribose) polymerase (PARP) [10]. The c-PARP protein is involved in cellular DNA damage repair and is cleaved during apoptosis. The MC-3 cells in this study showed that chrysin increased the Bax/Bcl-2 ratio and c-PARP levels. In previous studies, chrysin could increase Bax and decrease Bcl-2 in hepatocarcinoma cells, whereas c-PARP levels increased in thyroid cancer cells [26,27]. Thus, our results agree with those of previous studies, implying that chrysin induced apoptosis through the intrinsic pathway.

The increased AVO levels and the changes in proteins related to autophagy observed using acridine orange staining and Western blotting verified the ability of chrysin to induce autophagy in MC-3 cells. Previously, chrysin was shown to induce autophagy in colon cancer and endometrial cancer cells, in agreement with the results of this study [28,29]. In addition, the role of induced autophagy was confirmed using the autophagy inhibitors 3-MA (early-stage inhibitory) and HCQ (late-stage inhibitor). When electrophoretic autophagy was inhibited with 3-MA, no significant change in cell viability was observed. However, when late autophagy was inhibited with HCQ, cell viability decreased, suggesting that the cells exhibited a protective effect through apoptosis inhibition. In melanoma cells treated with dieckol along with 3-MA and HCQ, 3-MA was shown to reduce apoptosis while HCQ increased it [30]. Therefore, it can be inferred that autophagy causes cell protection in the early phase and apoptosis in the latter phase. This is contrary to the results obtained in our study, and this discrepancy can be attributed to the differences in cell types and stimuli. In conclusion, in the case of cancer cells treated with chrysin, inhibition of autophagy in the late stage rather than in the early stage can effectuate a greater anticancer effect.

The MAPK pathway is observed in various apoptotic cancer cells and is the main target of anticancer treatment [31]. In this study, groups treated with varying concentrations of chrysin showed decreased p-ERK levels and increased p-JNK and p-p38 levels, compared with those in the control. In addition, simultaneous treatment with chrysin and the ERK inhibitor PD98059 increased cell viability and apoptosis, compared with cell viability and apoptosis in the chrysin-only group. According to a previous study, when glioblastoma cells were treated with chrysin, cell proliferation decreased due to reduced p-ERK, and apoptosis was detected [32]. This coincided with the results of our study and suggested that chrysin-induced MC-3 cell apoptosis occurred via the MAPK/ERK pathway.

Autophagy is regulated by various signaling pathways, and mTOR, in particular, is known as a key regulator [33]. Notably, mTOR activation via the PI3K/AKT and MAPK/ERK pathways prevents excessive autophagy [17]. In this study, chrysin-mediated ERK reduction increased autophagy. In addition, additional treatment with the ERK inhibitor PD98059 increased autophagy, compared with autophagy after chrysin treatment alone, implying that the chrysin-increased autophagy also occurred via the MAPK/ERK pathway. However, further studies need to investigate the potential role of the PI3K/AKT pathway, another key pathway of autophagy.

Chrysin significantly reduced MC-3 cell viability by inducing apoptosis. Western blot analysis showed that chrysin increased the levels of the pro-apoptotic proteins c-PARP and Bax, while the level of Bcl-2, an anti-apoptotic protein, decreased. Furthermore, chrysin-induced apoptosis was shown to be mediated by the MAPK/ERK pathway. The ability of chrysin to induce autophagy in MC-3 cells was verified based on the changes in autophagy-related proteins (LC3-Ⅱ, Beclin-1, p62, and mTOR) and the increase in AVOs. MTT and flow cytometry assays using 3-MA and HCQ inhibitors showed that chrysin-induced autophagy exerts a cytoprotective effect via the ERK/mTOR pathway.

The findings in this study collectively suggest that chrysin exerts an anticancer effect against MC-3 cells, which can be enhanced by inhibiting autophagy in the late stage. However, further studies should be conducted to investigate whether chrysin can mediate apoptosis and autophagy in vivo, as well as the potential association between these two biological mechanisms.

## 4. Materials and Methods

### 4.1. Cell Lines and Reagents

MC-3 (Human MEC) cells were provided by Prof. Wu Junzheng (Fourth Military Medical University, Xi’an, China). The cells were cultured in Dulbecco’s modified Eagle’s medium (DMEM) supplemented with fetal bovine serum (FBS) (Welgene, Gyeongsan, Korea) and streptomycin/penicillin (Gibco BRL, Grand Island, NY, USA). Chrysin and the general reagents used in this study were purchased from Sigma-Aldrich Co. (St. Louis, MO, USA). Chrysin was dissolved in dimethyl sulfoxide (DMSO) for further use. A fluorescein isothiocyanate (FITC) Annexin V Apoptosis Detection Kit was purchased from BD Pharmingen (San Diego, CA, USA). Bax, PARP, LC3B, Beclin-1, p-ERK, p-JNK, p-p38, and anti-rabbit IgG secondary antibodies were purchased from Cell Signaling Technology (Danvers, MA, USA). Bcl-2, β-actin, and mouse IgG secondary antibodies were purchased from Santa Cruz Biotechnology Inc. (Dallas, TX, USA). HCQ and 3-MA were purchased from Selleck Chemicals (Houston, TX, USA). The ERK inhibitor PD98059 was purchased from ENZO Life Sciences Inc. (Oyster Bay, NY, USA).

### 4.2. Cell Culture

MC-3 oral cancer cells were cultured in DMEM supplemented with 5% FBS and 1% streptomycin/penicillin in an incubator maintained at 37 °C at 5% CO_2_. Cells were subcultured once they reached 80–90% confluence, and the medium was replaced every 2–3 days.

### 4.3. MTT Assay

Cell viability was assessed using the MTT assay. MC-3 cells were seeded into 96-well plates in DMEM medium (2 × 10^4^ cells/mL) and incubated at 37 °C in the presence of 5% CO_2_ for 24 h. Thereafter, the cells were treated with chrysin at varying concentrations (0, 25, 50, 75, and 100 µM) for 24 h. PD98059 (25 µM), HCQ (20 µM), and 3-MA (1 mM) were pre-treated for 3 h before chrysin treatment. Next, 40 µL of MTT solution dissolved in phosphate-buffered saline (PBS; 1 mg/mL) was added to each well, and the plate was incubated for 2 h. The culture medium was then removed, and 100 µL of DMSO was added to each well to fully dissolve the formazan crystals. Absorbance was measured at 595 nm using an ELISA reader (Bio-Rad Laboratories Inc., Hercules, CA, USA).

### 4.4. DAPI Staining

Cellular morphological changes induced by apoptosis were examined using DAPI staining. MC-3 cells were seeded in a 60 mm dish at a density of 1 × 10^5^ cells/mL for 24 h. Next, the cells were treated with chrysin (0, 50, and 100 µM) for another 24 h. Afterwards, the cells were fixed with 4% paraformaldehyde solution for 15 min, followed by treatment with DAPI solution, and they were then observed under a fluorescent microscope (Thornwood, NY, USA) at 200× magnification.

### 4.5. Annexin V-PI Staining

The apoptosis rate was measured using a FITC-Annexin V Apoptosis Detection Kit (San Diego, CA, USA). First, MC-3 cells were cultured in a 75 cm^2^ flask for 24 h. For annexin V/PI staining, MC-3 cells were treated with chrysin (0, 50, and 100 µM) for 24 h. Next, the cells were washed with PBS and harvested using trypsin-ethylenediaminetetraacetic acid (EDTA). The harvested cells were suspended in 1× binding buffer at 1 × 10^5^/mL. The annexin V staining kit (BD Pharmingen™, San Diego, CA, USA) was used for 20 min to stain the cells with FITC-conjugated annexin V and phycoerythrin-conjugated PI. The BD FAC Suite (v1.0.6, BD Life Sciences, Franklin Lakes, NJ, USA) was used for the analysis.

### 4.6. Acridine Orange Staining

Acidic vesicular organelles, a morphological feature of autophagy, were analyzed via acridine orange staining. MC-3 cells were seeded in a 60 mm dish at a density of 2 × 10^5^ cells/mL for 24 h. Next, the cells were treated with chrysin (0, 50, and 100 µM) for another 24 h. Afterwards, all media containing chrysin were removed, and the cells were washed with PBS three times. The cells were then fixed with 4% formaldehyde for 15 min. In order to remove the formaldehyde, the cells were washed with PBS three times. Finally, after treatment with acridine orange solution (5 µg/mL, 10 min), the cells were observed under a fluorescence microscope at 200× magnification.

### 4.7. Western Blot Analysis

Western blotting was performed to analyze the expression of apoptosis-related proteins. MC-3 oral cancer cells were cultured in a 75 cm^2^ flask at 1 × 10^6^ cells/mL for 24 h and then treated with chrysin (0, 50, and 100 µM). After 24 h, cells were harvested using trypsin-EDTA. A cell lysis buffer was added to the harvested cells for 20 min at 4 °C, and the mixture was centrifuged (15,920× *g*, 5 min, 4 °C) to obtain the supernatant as the cell lysate. Bradford protein analysis (Bio-Rad Laboratories) was used to quantify the concentration of the extracted proteins. The cell lysate was separated using 12% sodium dodecyl sulfate polyacrylamide gel electrophoresis, then transferred to nitrocellulose membranes (Hercules, CA, USA). After 1 h blocking of the membranes in 5% skim milk-TBST (20 mM Tris HCl, pH 7.5, 150 mM NaCl, 0.1% Tween 20), the target primary antibody (1:1000) was added for an overnight (4 °C) reaction. Next, rabbit IgG or mouse IgG (1:2000) was added for a 1 h reaction (22–23 °C). Afterwards, ECL detection reagents were used for blotting, while density was measured using Image J Launcher (provided by NCBI).

### 4.8. Statistical Analyses

All experiments were conducted in triplicate. The results are expressed as means ± S.D. The variation in mean values between the control and chrysin treatment groups was analyzed using one-way analysis of variance with Dunnett’s *t*-test. The level of significance was set at *p* < 0.05.

## Figures and Tables

**Figure 1 ijms-23-15747-f001:**
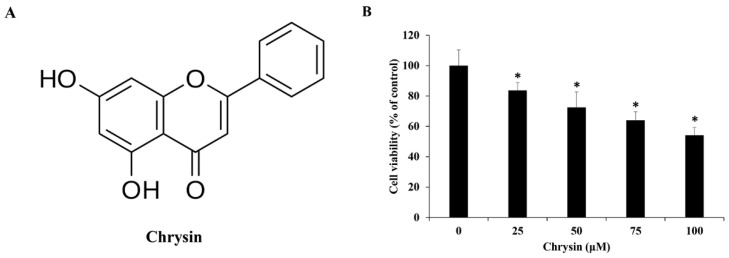
(**A**) The chemical structure of chrysin. (**B**) Chrysin reduces the viability of MC-3 oral cancer cells. The results are presented as the means and standard deviation of three independent samples. Statistical differences were determined by the Dunnett’s *t*-test: * *p* < 0.05, compared with the control.

**Figure 2 ijms-23-15747-f002:**
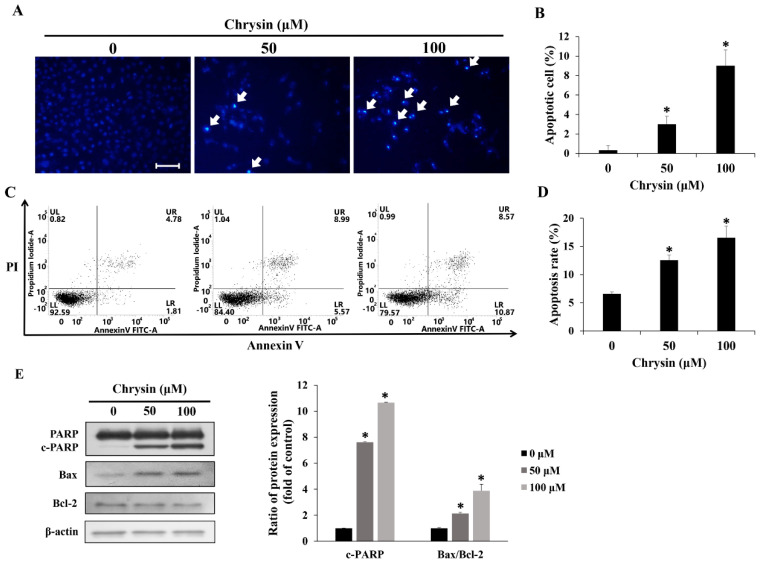
Chrysin induces apoptosis in MC-3 oral cancer cells. (**A**) Fluorescence microscopy image of cell morphology (scale bar = 10 μm) after chrysin treatment. The arrows indicate chromatin condensation in MC-3 cells. (**B**) The bar graph representing the average number of 4′,6-diamidino-2-phenylindole-positive cells as a percentage of all cells. (**C**) Cells stained with annexin V-propidium iodide and analyzed using flow cytometry. (**D**) Bar graph representing the percentage of cells in apoptosis among all cells. (**E**) The levels of the apoptosis-related proteins poly (ADP-ribose) polymerase, Bax, and B-cell lymphoma-2 (Bcl-2). Quantification was performed using Image J. The results are presented as the means and standard deviation of three independent samples. The statistical differences were determined by the Dunnett’s *t*-test: * *p* < 0.05, compared with the control.

**Figure 3 ijms-23-15747-f003:**
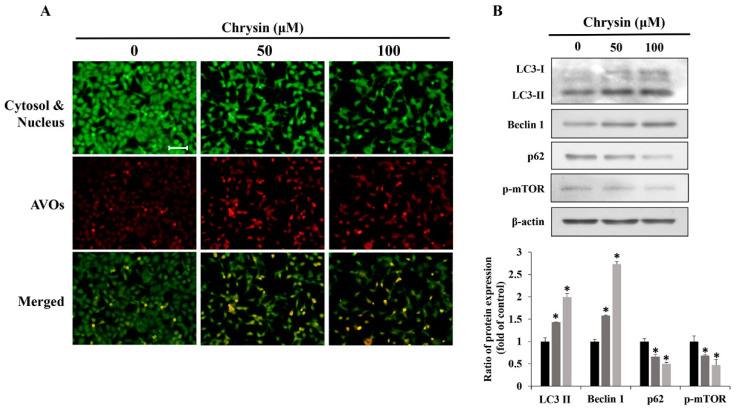
Chrysin induces autophagy in MC-3 oral cancer cells. (**A**) Fluorescence microscopy image of acidic vesicular organelles after chrysin treatment. Scale bar = 10 μm. The cytosol and the nucleus are stained fluorescent green, and the AVOs are stained fluorescent red. (**B**) The levels of the autophagy-related proteins light chain 3, B-cell lymphoma-2-interacting protein (Beclin-1), p62 and p-mammalian target of rapamycin (mTOR). Quantification was performed using Image J. The results are presented as the means and standard deviation of three independent samples. The statistical differences were determined by the Dunnett’s *t*-test: * *p* < 0.05, compared with the control.

**Figure 4 ijms-23-15747-f004:**
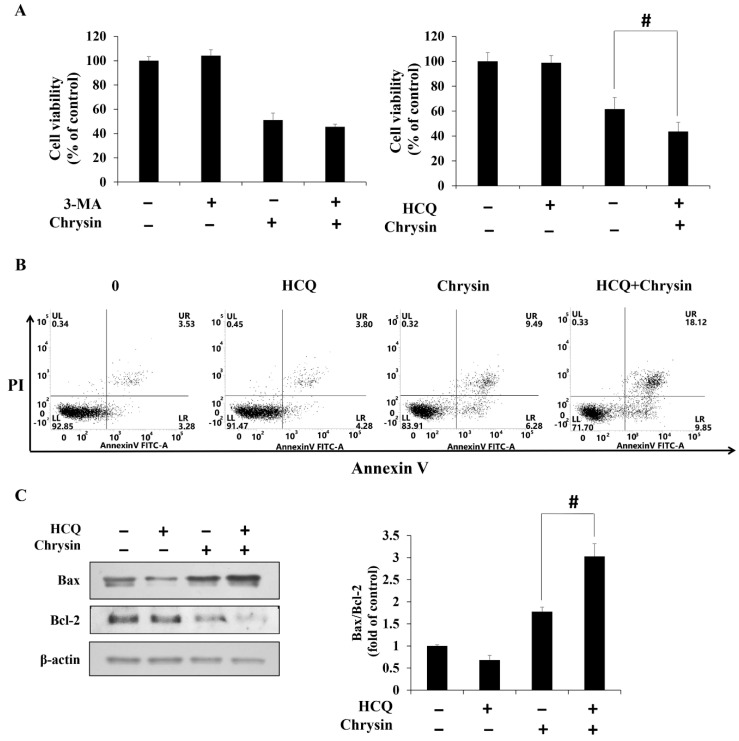
Chrysin induces cell protective autophagy in MC-3 oral cancer cells. (**A**) Viability of cells pre-treated with 3-methyladenine and hydroxychloroquine (HCQ) and subsequently treated with chrysin. (**B**) Apoptosis rates of cells pre-treated with HCQ and subsequently treated with chrysin. (**C**) The level of apoptosis-related proteins Bax and Bcl-2. Quantification was performed using Image J. The results are presented as the means and standard deviation of three independent samples. The statistical differences were determined by the Dunnett’s *t*-test: # *p* < 0.05, compared with the chrysin treatment group.

**Figure 5 ijms-23-15747-f005:**
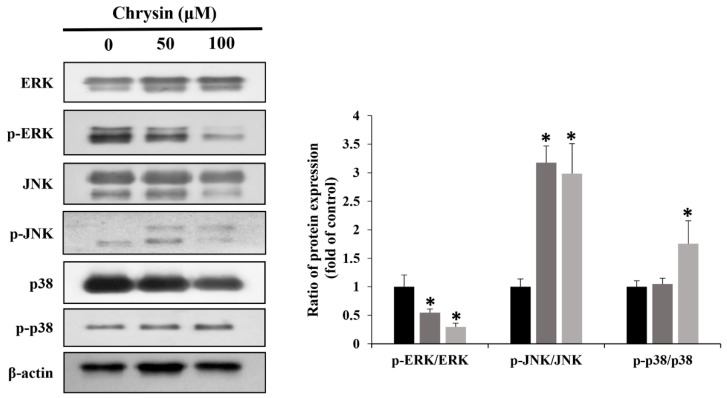
Chrysin induces apoptosis through mitogen-activated protein kinases [extracellular signal-regulated kinase (ERK), stress-activated protein kinase, and p38] in MC-3 oral cancer cells. Quantification was performed using Image J. The results are presented as the means and standard deviation of three independent samples. The statistical differences were determined by the Dunnett’s *t*-test: * *p* < 0.05, compared with the control.

**Figure 6 ijms-23-15747-f006:**
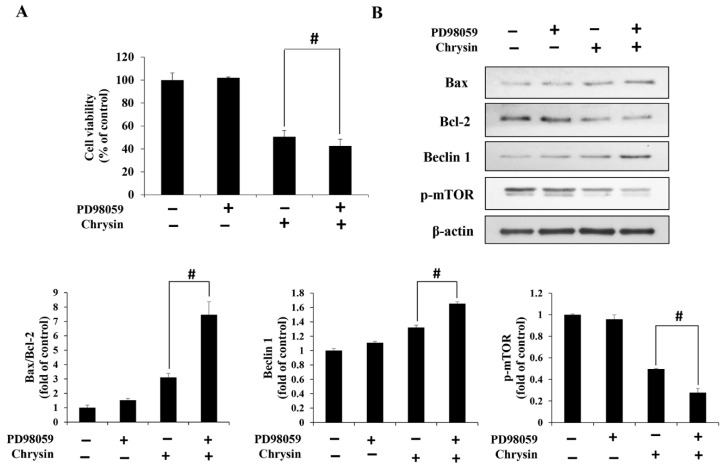
Chrysin induces autophagy through the ERK/mTOR pathway in MC-3 oral cancer cells. (**A**) The viability of cells pre-treated with PD98059 (an ERK inhibitor) and subsequently treated with chrysin. (**B**) The levels of the apoptosis- and autophagy-related proteins Bax, Bcl-2, Beclin-1, and p-mTOR. Quantification was performed using Image J. The results are presented as the means and standard deviation of three independent samples. The statistical differences were determined by the Dunnett’s *t*-test: # *p* < 0.05, compared with the chrysin treatment group.

## Data Availability

The datasets used and/or analyzed during the current study are available from the corresponding author on reasonable request.

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
