# Peer review of "Chrysin Induces Apoptosis via the MAPK Pathway and Regulates ERK/mTOR-Mediated Autophagy in MC-3 Cells"

_ijms, 2022, doi:10.3390/ijms232415747_

Round 1

Reviewer 1 Report

The proposed article is novel and interesting. The methodology is well described, the results well presented, and the discussion is appropriate. However, these very positive results would need to be complemented with more cell lines to corroborate Chrysin's antitumor activity, or to test it in non-tumor cells to demonstrate its selectivity for tumor cells.

Author Response

Response to Reviewer 1 Comments

Point 1: The proposed article is novel and interesting. The methodology is well described, the results well presented, and the discussion is appropriate. However, these very positive results would need to be complemented with more cell lines to corroborate Chrysin's antitumor activity, or to test it in non-tumor cells to demonstrate its selectivity for tumor cells.

Response: We greatly appreciated the reviewer’s efforts to carefully review the paper. Through previous papers, we applied chrysin within a concentration that does not reduce cell viability in normal cells. Nevertheless, it is considered necessary to add cell lines to confirm that the positive effect of chrysin also works in other cell lines, so there are plans to conduct additional research with several cell lines in the future.

Xue, C., Chen, Y., Hu, D. N., Iacob, C., Lu, C., & Huang, Z. (2016). Chrysin induces cell apoptosis in human uveal melanoma cells via intrinsic apoptosis. Oncology letters12(6), 4813-4820.

Reviewer 2 Report

This study reported the apoptotic effect of chrysin on MC-3 cells. And the authors showed that chrysin palyed roles in anti-cancer by inducing apoptosis via MAPK pathway and by evoking autophagy through ERK-mTOR pathway. The study is with great clinical significance, however there are major concerns that seriously limit the relevance of the study. Generally speaking, the result can not support the conclusion on the mechanistic insights. Specific comments to the authors are as follows: 

1. In 2.4 section of the main text, the authors described ...of autophagy marker proteins....p-mTOR decreased.  p-mTOR is related to autophagy, but not a marker of autophagy. The authors should clarify this in the manuscript.

2. In section 2.4, the authors only displayed the results of cell viability of 3-MA. I suggest the authors also show the apoptosis results of 3-MA.

3. HCQ and 3-MA showed different effects on MC-3 cells, I suggest the authors discuss this in the Discussion section.

4. In 2.5 section, the authors found that p-ERK1/2 decreased whereas the phosphorylations of JNK and p38 increased when treated with chrysin. In other words, when MC-3 cells were treated with chrysin, ERK MAPK pathway was suppressed, whiled JNL and p38 MAPK was activated. I dont understand why the authors further inhibited ERK by its inhibitor PD98059, not activated ERK, to investigate the effects of ERK MAPK pathway on cell viability and apoptosis in the following experiments. I think the strategy is incorrect.

5. Furthermore, the author did not study the effects of JNK and p38 pathways, and arbitrarily concluded that chrysin induced autophagy via ERK-mTOR pathway. The results can not support the conclusion.

6. In the section of 2.2. Chrysin induces morphological changes in MC-3 cells, the authors should describe the results of morphological changes.

Round 2

Reviewer 1 Report

The authors have made the proposed changes correctly.

Author Response

Thank you for the reviewer's good assessment.

Reviewer 2 Report

Most of y concerns have been addressed, however I dont think the authors have clearly clarified the mechanism of chrysin-induced autophagy. The authors should perform more experiments to demonstrate chrysin induces autophagy through ERK pathway, not via JNK or p38.

Author Response

Thanks for the good comments from the reviewer. We agree with the reviewer's opinion and believe that additional experiments on JNK and P38 pathways are necessary to prove a clear ERK pathway. In future studies, we plan to more clearly identify the autophagy pathway by observing other pathways in addition to the target pathway.